# Polycomb mutant partially suppresses DNA hypomethylation–associated phenotypes in Arabidopsis

Martin Rougée[1,*], Leandro Quadrana[2,*], Jérôme Zervudacki[2,*], Valentin Hure[1], Vincent Colot[2] ⓘ, Lionel Navarro[2] ⓘ, Angélique Deleris[1] ⓘ

In plants and mammals, DNA methylation and histone H3 lysine 27 trimethylation (H3K27me3), which is deposited by the polycomb repressive complex 2, are considered as two specialized systems for the epigenetic silencing of transposable element (TE) and genes, respectively. Nevertheless, many TE sequences acquire H3K27me3 when DNA methylation is lost. Here, we show in *Arabidopsis thaliana* that the gain of H3K27me3 observed at hundreds of TEs in the *ddm1* mutant defective in the maintenance of DNA methylation, essentially depends on CURLY LEAF (CLF), one of two partially redundant H3K27 methyltransferases active in vegetative tissues. Surprisingly, the complete loss of H3K27me3 in *ddm1 clf* double mutant plants was not associated with further reactivation of TE expression nor with a burst of transposition. Instead, *ddm1 clf* plants exhibited less activated TEs, and a chromatin recompaction as well as hypermethylation of linker DNA compared with *ddm1*. Thus, a mutation in polycomb repressive complex 2 does not aggravate the molecular phenotypes linked to *ddm1* but instead partially suppresses them, challenging our assumptions of the relationship between two conserved epigenetic silencing pathways.

## Introduction

DNA methylation is an epigenetic mark involved in the stable silencing of transposable elements (TEs) as well as the regulation of gene expression in plants and mammals. When present over TE sequences, it is usually associated with the di- or trimethylation of histone H3 lysine 9 (H3K9me2/H3K9me3) and, in plants, positive feedback loops between the two marks exist, which maintain TE sequences in the heterochromatic state (Johnson et al, 2002; Du et al, 2015). In *Arabidopsis thaliana*, DOMAINS REARRANGED METHYLASE 2 (DRM2) establishes DNA methylation in all three sequences contexts

(i.e., CG, CHG, and CHH) in a pathway referred to as RNA-directed DNA methylation (RdDM) that involves small RNAs (Cao & Jacobsen, 2002; (Chan et al, 2004). Maintenance of DNA methylation over TEs is achieved by the combined and context-specific action of DRM2-RdDM (CHH methylation), CHROMOMETHYLASES 2 and 3 (CMT2 and CMT3, for CHH and CHG methylation, respectively) (Zemach et al, 2013; Stroud et al, 2014), and METHYLTRANSFER-ASE1 (MET1) (CG methylation) (Kankel et al, 2003). In addition, the SNF2 family chromatin remodeler DECREASE IN DNA METHYLA-TION 1 (DDM1) is necessary for DNA methylation of most heterochromatic sequences in all cytosine sequence contexts (Stroud et al, 2013; Zemach et al, 2013).

In contrast to DNA methylation, histone H3 lysine 27 trimethylation (H3K27me3), which is targeted by the highly conserved polycomb group (PcG) proteins, in particular polycomb repressive complex 2 (PRC2), is a hallmark of transcriptional repression of protein-coding and microRNA genes in plants as well as in animals (Prete et al, 2015; Förderer et al, 2016; Chica et al, 2017; Marasca et al, 2018); it is thought to act by promoting a local compaction of the chromatin that antagonizes the transcription machinery (Prete et al, 2015) to maintain transcriptional silencing (Holoch & Margueron, 2017). Thus, H3K27me3 and DNA methylation are generally considered as mutually exclusive chromatin marks.

Nonetheless, there is a growing body of evidence of an interplay between the two silencing pathways. In particular, in both plants and mammals, many TE sequences gain H3K27me3 upon their loss of DNA methylation (Mathieu et al, 2005; Deleris et al, 2012; Reddington et al, 2013; Saksouk et al, 2014). Moreover, in the filamentous Neurospora, H3K27me3 is redistributed from gene to TE-rich constitutive heterochromatin when the heterochromatic mark H3K9me3 or the protein complexes that bind to it are lost (Basenko et al, 2015).

The fact that H3K27me3 can mark TE sequences upon their demethylation led to the idea that PcG could serve as a back-up silencing system for hypomethylated TEs (Deleris et al, 2012). Consistent with this notion, subsequent work in mammals showed that H3K27me3 re-established the repression of thousands of

[1]Université Paris-Saclay, Commissariat à l'Énergie Atomique et aux Énergies Alternatives (CEA), Centre National de la Recherche Scientifique (CNRS), Institute for Integrative Biology of the Cell (I2BC), Gif-sur-Yvette, France  [2]Institut de Biologie de l'Ecole Normale Supérieure (IBENS), Ecole Normale Supérieure, CNRS, Institut National de la Santé et de la Recherche Médicale (INSERM), Paris, Sciences et Lettres (PSL) Research University, Paris, France

Correspondence: angelique.deleris@i2bc.paris-saclay.fr
*Martin Rougée, Leandro Quadrana, and Jérôme Zervudacki contributed equally to this work

hypomethylated TEs in embryonic stem cells subjected to rapid and extensive DNA demethylation (Walter et al, 2016).

In a previous study, we provided evidence supporting a role of PcG in the transcriptional silencing of *EVADE* (*EVD*) (Zervudacki et al, 2018), an *A. thaliana* retroelement of the *ATCOPIA93* family that is tightly controlled by DNA methylation and which transposes in plants mutated for the chromatin remodeler *DDM1* (Tsukahara et al, 2009). We observed that silencing of *EVD* is dependent on both DNA methylation and H3K27me3, which, at this locus, depends on the SET-domain protein CURLY LEAF (CLF) (Zervudacki et al, 2018). Whether the dual control observed at *EVD* is also present at other plant TEs is unknown.

In the present work, we integrated genetics, epigenomics, and cell imaging to show that numerous TEs gain H3K27me3 in response to *ddm1* mutation-induced loss of DNA methylation. We demonstrate also that this gain is mediated by CLF, with no apparent role for the SET-domain H3K27 methyltransferase SWINGER (SWN), which is also active in vegetative tissues and otherwise partially redundant with CLF (Chanvivattana et al, 2004; Wang et al, 2016; Yang et al, 2017). Unexpectedly, the combination of *ddm1* and *clf* mutations was not associated with further reactivation of TE expression or transposition as compared with *ddm1*, except for *EVD*. Instead less TEs were expressed, and we observed a partial recompaction of heterochromatin coupled with DNA hypermethylation, prominently in linker DNA, in *ddm1 clf* versus *ddm1*. Together, these results show that globally altering PRC2 activity partially suppresses *ddm1* phenotypes.

# Results

### Hundreds of hypomethylated TEs gain H3K27me3 in *ddm1*

We previously showed that in *met1* mutants impaired for CG methylation, hundreds of TEs gain H3K27me3 methylation in Arabidopsis (Deleris et al, 2012). Whereas *met1* and *ddm1* mutants are both globally hypomethylated, the later, in contrast to the former, affects almost exclusively TE and other repeat sequences, and in the three sequence contexts. Thus, the *ddm1* mutant appeared to be a more relevant background to directly explore the interplay between DNA methylation and Polycomb at TE sequences, and we conducted an H3K27me3 ChIP-seq experiment in this mutant. Like in *met1*, hundreds of TEs showed increased accumulation of H3K27me3 in *ddm1* (Fig 1A–C). A subset of 672 TEs showed no H3K27me3 ChIP-seq signal in wild-type plants and significantly gained H3K27me3 over their full length in *ddm1* (Fig 1D). Moreover, the vast majority of those 672 TEs are located in pericentromeric regions (Fig 1E) and were included in the subset of TEs that gain H3K27me3 in *met1* (Deleris et al, 2012) (Fig 1F) possibly because the extent of TE hypomethylation in *ddm1* is less than in *met1*, where all CG methylation (the most abundant) is virtually eliminated (Stroud et al, 2013). This argues for a major role of DNA methylation, in particular CG methylation, and rather than non-CG methylation associated with H3K9me2, in antagonizing PRC2 as previously proposed (Mathieu et al, 2005; Deleris et al, 2012).

Two major TE super families (LTR/Gypsy, DNA/others) were overrepresented among the 672 TEs that significantly gain H3K27me3 in ddm1 as compared with the distribution of the heterochromatic, pericentromeric TEs (targets of DDM1) families (Fig S1A). The differences in TE-type targeting between *met1* and *ddm1* (Fig S1A) likely reflect a differential sensitivity of the TE families to the different mutations as for DNA methylation, thus the differential extent of TE hypomethylation and loss of PRC2 antagonism by DNA methylation as discussed earlier. In addition, the overrepresentation of two TE families among the TEs that gain H3K27me3, common to both mutants (Fig S1A) could suggest the existence of sequence-specific targeting. Indeed, PRC2 can be targeted to specific genes by the recognition of short sequence motifs (Xiao et al, 2017). Consistent with this motif-based targeting, we detected a significant enrichment of the Telobox, CTCC, GA-repeat, and AC-rich motifs in the 672 TEs compared with the rest of the heterochromatic TEs (Figs 1G and S1B). These results suggest the presence of an instructive mechanism of PRC2 recruitment at TEs with particular motifs used as nucleation sites either through direct sequence recognition or indirectly, through chromatin structures that could be promoted by these sequences. In addition, the detection of continuous blocks of H3K27me3, in particular on chromosomes 1 and 4 (Fig S1C), may suggest that, once nucleated, H3K27me3 domains spread over entire TE sequences and even beyond, into nearby TEs, in a fashion similar to what was described at genic sequences (Wang et al, 2016; Yang et al, 2017). Of note, ectopic gain of H3K27me3 to TEs in *ddm1* did not seem associated with a loss of H3K27me3 at genes (Fig S1D) and no gene lost H3K27me3 significantly in ddm1 in our differential analysis, contrary to what was observed in *met1* (Deleris et al, 2012). In the scenario whereby the gain of H3K27me3 at TEs would be the result of redistribution from genes to TEs, this could be explained by the lesser number of TEs targeted by PRC2 in *ddm1* versus *met1* (Fig 1F). Alternatively, or in addition, loss of H3K27me3 at genes in *met1* but not in *ddm1* could be contributed by the pronounced ectopic DNA hypermethylation at many genes, particularly H3K27me3-marked genes in *met1* (Deleris et al, 2012), which we did not detect globally in *ddm1* (Fig S1E) even if this phenomenon could occur sporadically at specific genic loci like AGAMOUS (Jacobsen et al, 2000).

### The gain of H3K7me3 over hypomethylated TEs depends on CLF

To test whether CLF is required for the gain of H3K7me3 over TEs, we performed a second set of ChIP-seq experiments in two different F3 progenies of *ddm1 clf* double mutant (and that we refer to as biological replicates, here BR1 and 2). Because total levels of H3K27me3 are strongly reduced in both *clf* and *ddm1 clf* mutants (Fig S2A), compared with wild type, we spiked-in exogenous Drosophila chromatin in the chromatin extracts for normalization (Orlando et al, 2014). There was no consistent difference in the levels of H3K27me3 at genes between *ddm1 clf* and *clf* mutant (Fig S2B), in accordance with DDM1 affecting TE and other repeat sequences specifically (Lippman et al, 2004). Conversely, the gain of H3K27me3 observed over TEs in *ddm1* was almost completely abolished in *ddm1 clf* (Figs 2A–C and S2C), while being unchanged at all TEs tested in *ddm1 swn* (Figs 2D and S2D). Together, these results show that deposition of H3K27me3 at most TEs in *ddm1* is fully dependent on *CLF* with no apparent role

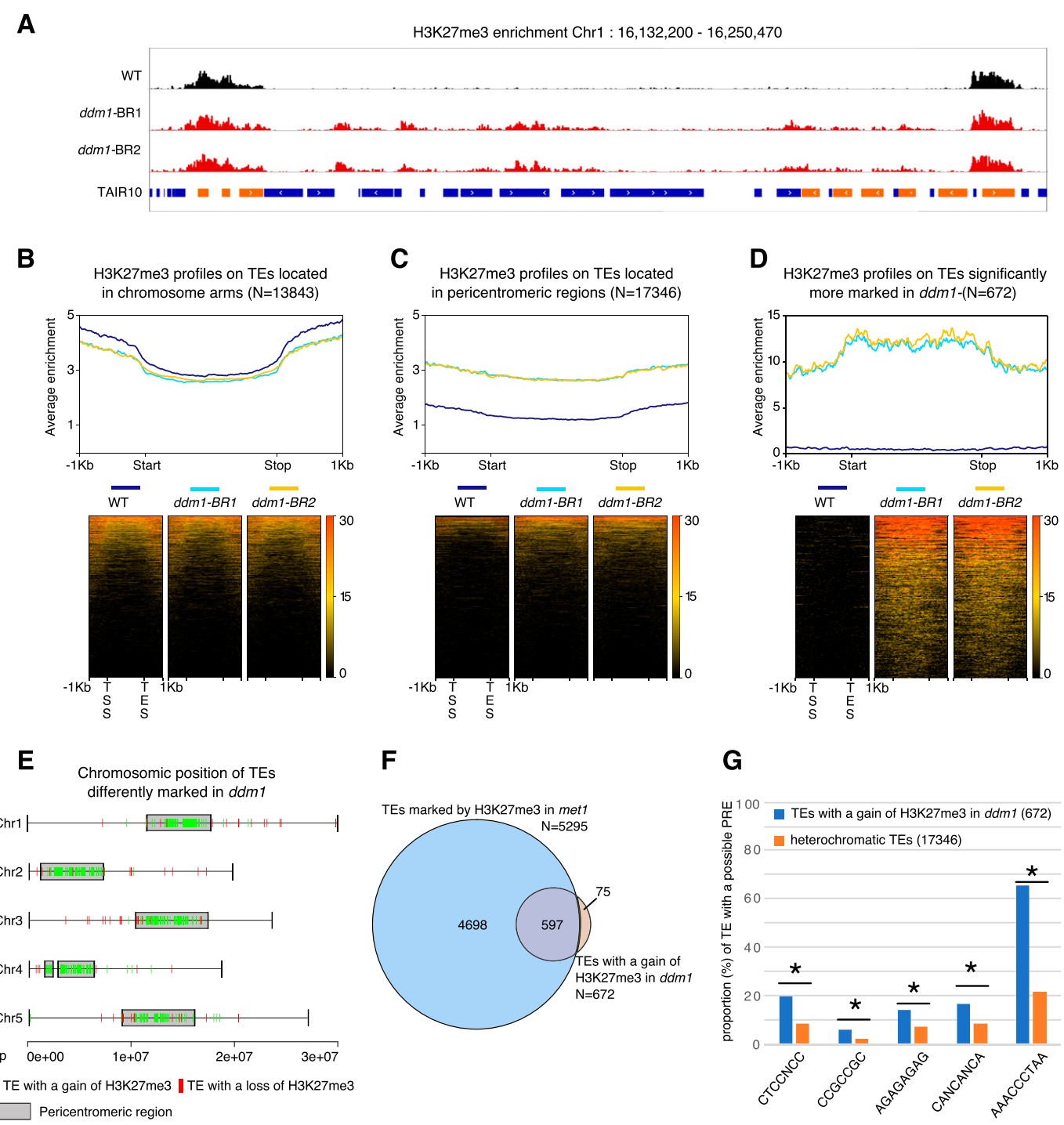

**Figure 1. A mutation in *DDM1* leads to a gain of H3K27me3 on some heterochromatic transposable elements (TEs).**

**(A)** Representative genome browser view of H3K27me3 levels in a 100 kb heterochromatic region of chromosome 1 in wild-type (WT) and two biological replicates (BR) of *ddm1* (blue bar: TE; orange bar: gene). **(B, C, D)** H3K27me3 enrichment in WT, *ddm1*-BR1, and *ddm1*-BR2 over TEs located in euchromatin (B), heterochromatin (C) and TEs with a significant positive fold change (FC) of H3K27me3 in *ddm1* (Pval < 0.1; Log2FC > 2) (D). The upper graph shows the mean enrichment of H3K27me3 over a given TE subset; the corresponding heat map below ranks them from top to bottom according to the average enrichment in all genotypes. **(E)** Chromosomal distribution of the 672 TEs that significantly gain H3K27me3 (Pval < 0.1; Log2FC > 2) and the 61 TEs that significantly lose H3K27me3 in *ddm1* (Pval < 0.1; Log2FC < −2). **(F)** Overlap between TEs marked by H3K27me3 in *met1* (Deleris et al, 2012, ChIP–ChIP data) and TEs with a significant gain of H3K27me3 in *ddm1* (ChIP-seq). **(G)** Graph showing the proportion of TEs (TEs with a gain of H3K27me3 in *ddm1* compared with heterochromatic TEs) with a given possible PRE as described in Xiao et al (2017) (* indicates a significant difference Pval < 0.5, *t* test). Source data are presented in Table S1.

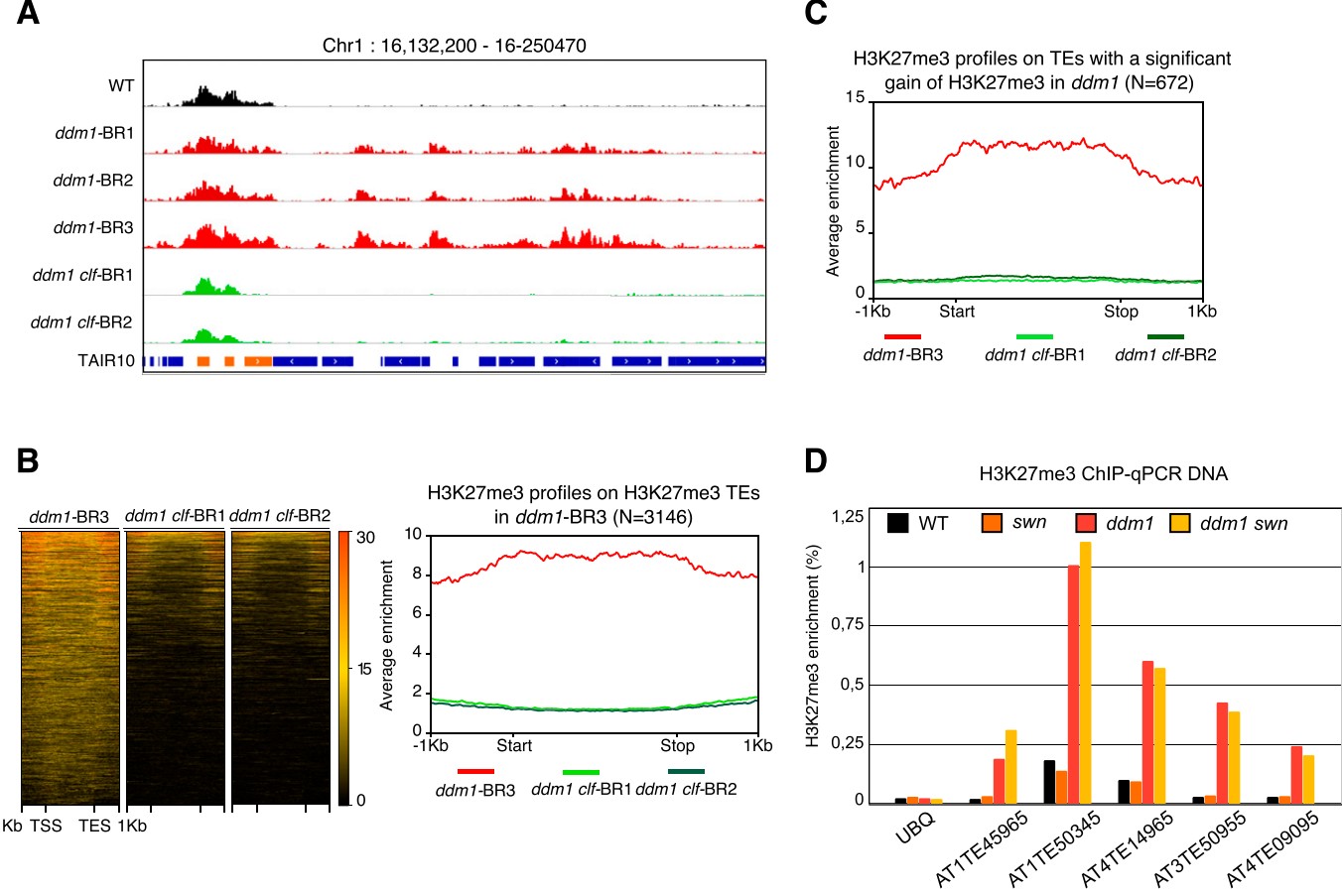

**Figure 2. The gain of H3K27me3 in *ddm1* is lost in *ddm1 clf*.**
**(A)** Genome browser view of H3K27me3 levels on the same region as Fig 1. **(B)** H3K27me3 enrichment in *ddm1* and two *ddm1 clf* replicates over transposable elements (TEs) with a H3K27me3 domain in *ddm1* with corresponding heat map ranking TEs according to the H3K27me3 mean value. **(C)** Metagene representing H3K27me3 enrichment in *ddm1*-BR3, *ddm1 clf*-BR1, and *ddm1 clf*-BR2 of the 672 TEs previously shown to gain H3K27me3 in ddm1. **(D)** ChIP-qPCR analysis of H3K27me3 marks in *ddm1 swn* double mutant at representative TEs that gain H3K27me3 marks in *ddm1*. Data were normalized to the input DNA, *UBIQUITIN* (*UBQ*) serves as a negative control. Because of technical variability in the ChIP efficiency between the two biological replicates, ChIP experiments are presented independently and a biological replicate is shown in Fig S2D. Source data are presented in Table S2.

of the paralogous histone methyltransferase *SWN*. This is in contrast with the well-established, partial dependency of H3K27me3 deposition at genes on CLF (Wang et al, 2016; Yang et al, 2017), which has been proposed to be due to a specialization of this factor in mediating amplification and spreading of H3K27me3 marks after their establishment by SWN (Yang et al, 2017). Thus, at least over *ddm1*-dependent hypomethylated TEs, CLF is required for the nucleation, and perhaps, additionally, spreading of H3K27me3.

## TE activation in *ddm1* is not enhanced in *ddm1 clf*

We previously described the phenotype of *ddm1 clf* mutants at the rosette stage (4.5 wk-old) in F2 plants (Zervudacki et al, 2018). In all mutant lines, we further observed floral defects (Fig S3A), severity of which seems to increase with generation time, resulting in almost complete sterility of F4 generation mutants. In addition, by examining the plants at the seedling stage (15-d-old), we could notice additional phenotypes in two of three independent *ddm1 clf* lines (chlorosis in one line and growth arrest in another) that segregated

1:3 and thus evoked the segregation of recessive mutations (Fig S3B). We previously showed that there is an enhanced accumulation of *ATCOPIA93* mRNAs in *ddm1 clf* double mutant as compared with *ddm1* (Zervudacki et al, 2018); thus, we hypothesized that these segregating phenotypes were caused by the transposition of TEs activated in *ddm1 clf*.

To test whether *ATCOPIA93* activity is increased and whether additional copies of this TE family accumulate in *ddm1 clf*, we performed Southern blot analysis (Fig 3A) using a probe specific of *ATCOPIA93 EVD/ATR* (*ATR* is an almost identical copy of *EVD*). We could only observe one copy of *EVD* and *ATR* in the wild type, *clf* single mutants as well as in the *ddm1* second generation inbred mutants. This result was not surprising because *EVD* was previously found to be active in *ddm1* or *ddm1*-derived epigenetically recombinant inbred lines, but this was observed in late generations (eighth and beyond) (Marí-Ordóñez et al, 2013; Tsukahara et al, 2009). By contrast, we observed linear extrachromosomal DNA in all the *ddm1 clf* double-mutant lines tested (progenies of individual F2), consistent with previous results (Zervudacki et al, 2018); in addition, in two of

three double-mutant lines, we detected numerous additional bands corresponding to *EVD/ATR* new insertions (Fig 3A). This indicates that transposition occurs in the double mutant, although to various extents which may reflect different dynamics of mobilization after the initial event (Quadrana et al, 2019). Pyrosequencing performed on the mutants showed that *ATCOPIA93* DNA (extrachromosomal and integrated) was mostly contributed by *EVD* (Fig 3B). Thus, DNA methylation and H3K27me3 act in synergy to negatively control *EVD* activity and prevent its transposition, in accordance with the dual control they exert on its transcription (Zervudacki et al, 2018). To test whether more TEs are activated in *ddm1 clf*, we performed TE-sequence capture (Baillie et al, 2011; Quadrana et al, 2016), which enables to detect with high sensitivity and specificity insertions that are present at a frequency as low as 1/1,000 within a DNA sample (Quadrana et al, 2019). Essentially, the capture probes cover 200 bp at each end of 310 potentially mobile TEs, which belong to 181 TE families either identified as mobile in various Arabidopsis ecotypes using the split-read approach, or for which non-degenerate and thus potentially mobile copies are present in the Col-0 genome (Quadrana et al, 2016). Our rationale was that the activity of some of these TEs is weak or absent in the Col-0 *ddm1* mutants or their derived epigenetic recombinant inbred lines because there is a second layer of silencing mediated by H3K27me3 in these epigenetic mutant backgrounds. Using about 100 seedlings of wild type, *ddm1* (second generation inbred mutants) and *ddm1 clf* (F3 progenies), no new insertions were detected for any of the 250 TEs tested in this manner, except for *EVD*, which had accumulated more copies in *ddm1 clf* than in *ddm1* (Fig 3C), although this difference was not statistically significant in average—this was presumably because the *ddm1 clf* line (#1), where only ecDNA accumulates, was included in the experiment along lines #2 and #3 (Fig S3C).

To investigate further the relationship between H3K27me3 and TE silencing, we performed RNA-seq experiments in three and four biological replicates (BR) of *ddm1* and *ddm1 clf* (four different F3 progenies), respectively. In all *ddm1* populations, about a 1,000 TEs (929) were up-regulated, the large majority of which (865) did not show a significant gain of H3K27me3 marks in this background. We found that a similar (*ddm1 clf*-BR3/4) or even lower (*ddm1 clf*-BR1/2) number of TEs were transcriptionally active in *ddm1 clf* compared with WT (Fig 3D) in keeping with the lack of enhanced transposition in *ddm1 clf*. In addition, these transcriptionally active TEs were expressed at similar levels (*ddm1 clf*-BR3/4) or less expressed (*ddm1 clf*-BR1/2) in *ddm1 clf* than in *ddm1* (Fig 3D). As for the TEs with a significant gain of H3K27me3 in *ddm1* (N = 672), we did not observe that they were more expressed upon complete loss of H3K27me3 in *ddm1 clf* than in *ddm1*: rather, they tended to be even less expressed than in *ddm1* in two *ddm1 clf* progenies (BR1/2) of four (Fig 3E). Accordingly, among these TEs with a significant gain of H3K27me3 in *ddm1*, we found only 15 TEs that were transcriptionally active in all *ddm1 clf* F3 progenies (Fig 3F). Thus, with the exception of *EVD* case study, which supports our initial hypothesis of a dual epigenetic control by DNA methylation and H3K27me3, it appears that *ddm1*-induced activation of TEs is not enhanced in *ddm1 clf* and, rather, tends to be partially suppressed in this double mutant background, with some heterogeneity between the lines used.

## *ddm1 clf* displays chromatin recompaction and increased DNA methylation compared with *ddm1*

To understand the mechanisms underlying the antagonism between DNA methylation and *CLF*-dependent H3K27me3 deposition, we performed cytogenetic analyses on nuclei from single and double mutants for *DDM1* and *CLF*. Chromocenters formation and DNA compaction were unchanged in *clf* but strongly impacted in *ddm1* as previously published (Soppe et al, 2002). By contrast, they were only partially affected in the double mutant, indicating that H3K27me3 loss induces chromatin recompaction specifically in the *ddm1* background (Fig 4A and B). In addition, immunofluorescence experiments showed that this chromatin recompaction was associated with the recovery of H3K9me2 distribution (Fig 4C).

To assess whether chromatin recompaction associates with DNA methylation gain, we compared the DNA methylomes of *ddm1* and *ddm1 clf* mutant plants. Consistent with the cytogenetic analyses, hundreds of loci exhibited higher DNA methylation in all three sequence contexts in *ddm1 clf* compared with *ddm1*, with gains being generally most pronounced at CHH sites and variable from one line to the other (Fig 5A).

To test whether H3K27me3 directly antagonizes DNA remethylation in *ddm1* (or promotes *ddm1*-induced DNA methylation loss) in *cis*, we searched for differential methylated regions (DMRs) using non-overlapping 100 bp windows. There were few hyper DMRs and these tended to be inconsistent across replicates, suggesting that they were the result of stochastic variation of DNA methylation (Becker et al, 2011). Moreover, some TEs showing clear DNA methylation gain in *ddm1 clf* compared with *ddm1* (an example is shown in Fig 5B) were not identified by our DMR detection. Visual inspection of these DNA hypermethylated TEs revealed that increased DNA methylation was in fact confined to short sequences (~20 bp), typically separated by ~150 bp (Fig 5B for an example of DNA hypermethylation in CG and CHG contexts, respectively; Fig S4A for all three contexts). Indeed, we identified 1208 TE-specific short (20 bp) sequences with higher CG methylation in *ddm1 clf* compared with *ddm1*. These small-size hyper-DMRs were distributed over 759 TEs, most of which (N = 571) are pericentromeric, and they were associated with a reduction of H3K27me3 levels (Fig 5C, H3K27me3 profiles centered on CG-DMRs). Among these 759 TEs hypermethylated in *ddm1 clf* compared with *ddm1*, 61 showed a significant gain of H3K27me3 in *ddm1*, representing 3.8 more enrichment ($P < 3.644 \times 10^{-19}$) compared with the number of TEs in the genome that would be both CG-hypermethylated (in *ddm1 clf*) and H3K27me3-marked (in *ddm1*) by chance. Although significant, this overlap is small, which suggests that, globally, the antagonism between H3K27me3 and DNA methylation is indirect. We did not find any consistent expression changes for the components involved in DNA methylation in *ddm1 clf* (Fig S4B), thus these observations cannot be explained, even partially, by the impact of *ddm1 clf* double mutation on the transcriptome. Besides, even if similar patterns of DNA hypermethylation compared with *ddm1* were previously observed in *ddm1 h1* where both DDM1 and canonical linker histone genes H1.1 and H1.2 are mutated (Zemach et al, 2013; Lyons & Zilberman, 2017), *ddm1 h1* mutant does not phenocopy the *ddm1 clf* mutant with regards to chromocenter formation: in fact, contrary to *ddm1 clf*, *ddm1 h1* did not induce DNA recompaction (Fig S4C) in agreement with the H1 role in chromatin condensation (He et al, 2019). Thus, the DNA hypermethylation observed in *ddm1 clf* versus *ddm1* cannot be attributed either to a histone H1 loss-of-function in this genetic background.

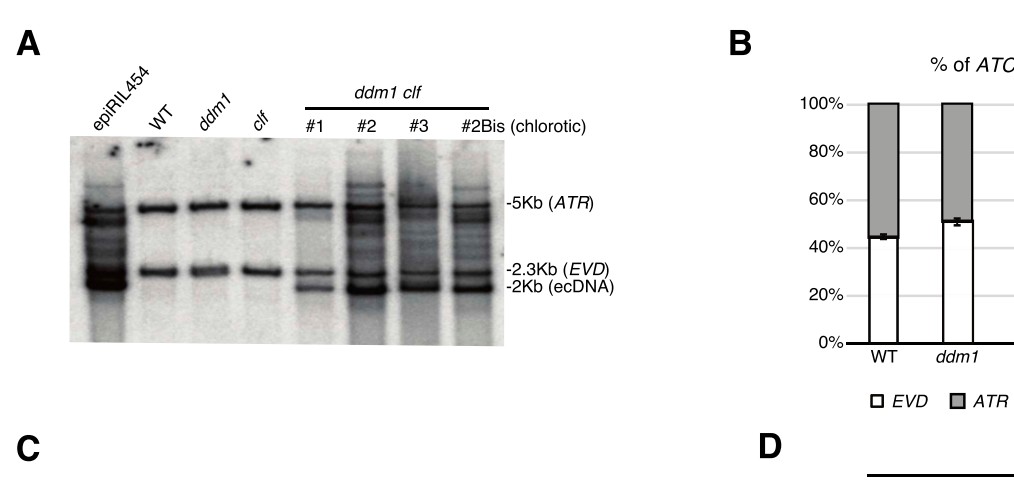

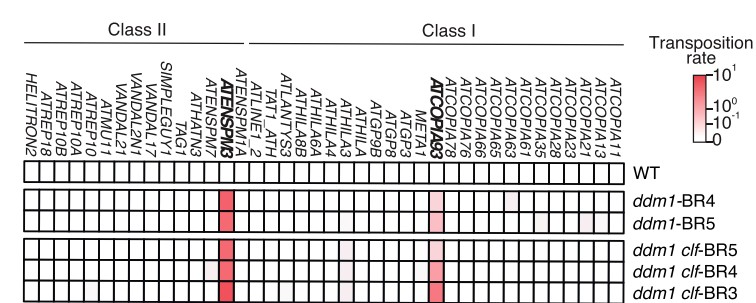

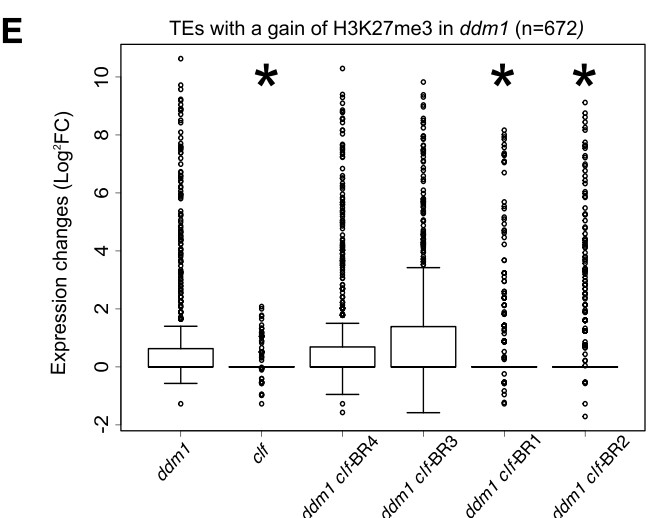

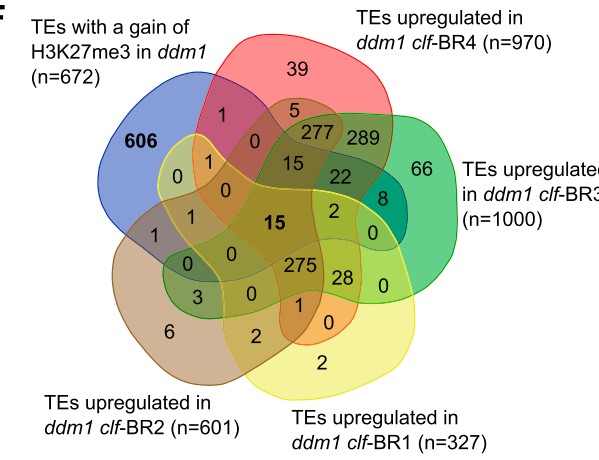

**Figure 3. Transposon activation in *ddm1* and *ddm1 clf*.**
**(A)** Southern blot analysis on a pool of leaves (eight rosettes for each genotype), using an *ATCOPIA93* probe that recognizes both *EVD* and *ATR* copies; epiRIL454 (Marí-Ordóñez et al 2013) was used as a positive control for activation of *ATCOPIA93* and three independent *ddm1 clf* lines were tested. **(B)** Qualitative analysis of genomic DNA by pyrosequencing. The position interrogated corresponds to the discriminating SNP between *EVD* (C/G) and *ATR* (A/T) and the % indicated represent the % of G (*EVD*, white bar) or T (*ATR*, grey bar). **(C)** Heat map showing the transposition rates of 40 potentially mobile transposable element (TE) families as determined by TE-capture (Quadrana et al, 2016) using 100 seedlings for each genotype. **(D)** Table showing up-regulated and down-regulated TEs compared with WT as identified by RNA-seq analyses on seedlings. **(E)** Box plot of log2 RNA fold changes (mutant/WT) over the TEs that significantly gain H3K27me3 in *ddm1*; * indicates a significant decrease in the fold change for a given *ddm1 clf* line relative to *ddm1* (Pval < 0.05, t test). **(F)** Venn diagram showing the overlap between TEs with a significant gain of H3K27me3 in *ddm1* and TEs up-regulated in the four different *ddm1 clf* lines. Source data are presented in Table S3.

Last, the apparent ~150 bp spacing between short-DMRs at discrete loci (Figs 5B and S4A), which is roughly the distance between the centers of consecutive nucleosomes, prompted us to test whether DNA hyper-methylation in *ddm1 clf* versus *ddm1* is periodic and if it occurs preferentially on non-nucleosomal linker DNA. We calculated the strength of DNA methylation periodicity with Fast Fourier transform calculation at CG hyper-DMRs within TEs (Chodavarapu et al, 2010; Lyons & Zilberman, 2017) (see the Materials and Methods section) and could show that CG methylation in *ddm1 clf* has indeed a strong periodicity of ~150 bp in relation to well-positioned nucleosomes (Fig 5D). Furthermore,

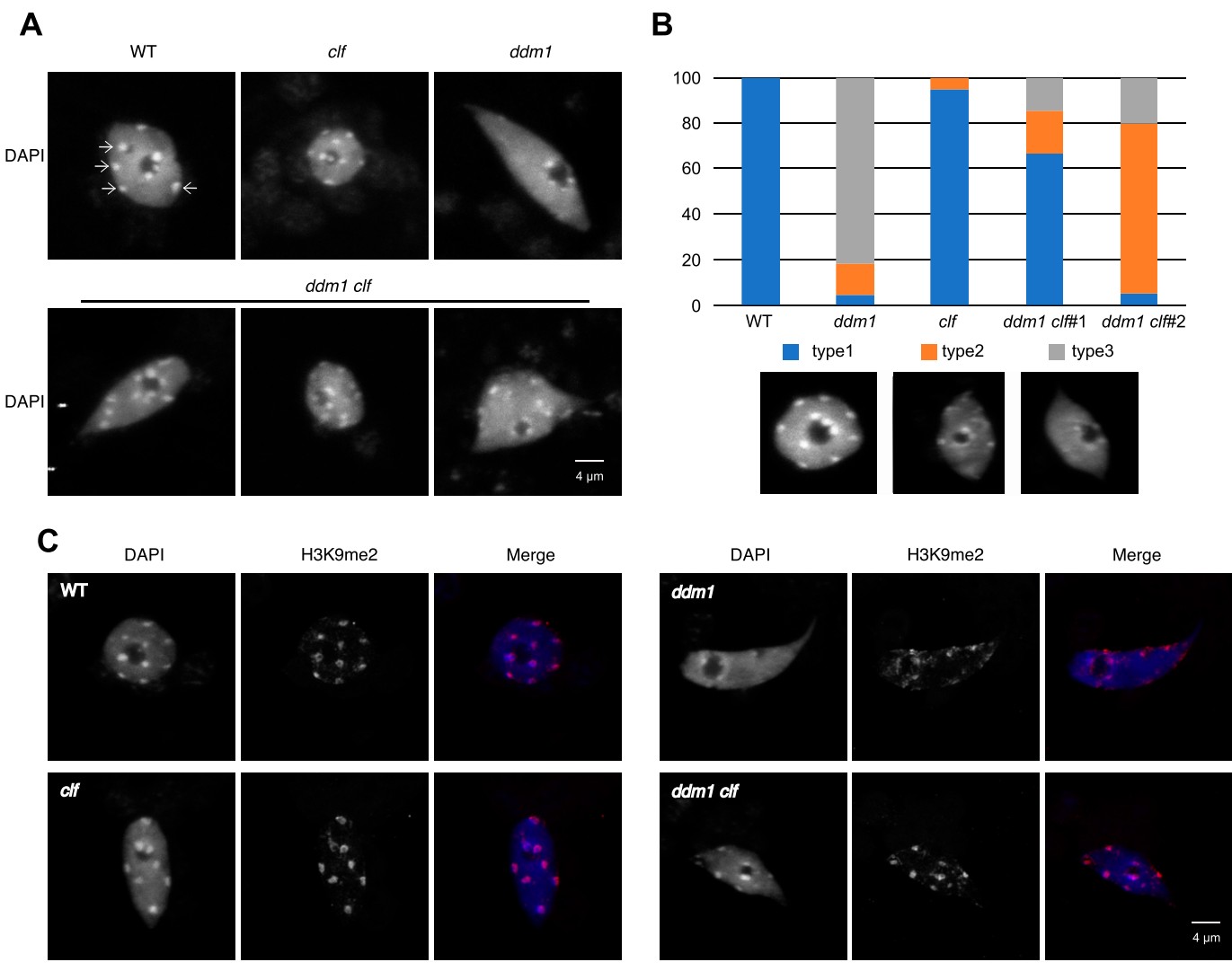

**Figure 4. The double mutant *ddm1 clf* displays a partial recompaction of DNA and more visible H3K9me2 compared with *ddm1*.**
**(A)** Representative pictures of DAPI stained nuclei extracted from 50 fixed 10 d-old Arabidopsis seedlings for each genotype (three independent F3 lines from *ddm1 clf*). **(B)** Nucleus types were blindly categorized according to the signal distribution relative to the heterochromatic chromocenters. Type1 compacted (like WT), type2; semi-compacted and type 3; decompacted (like *ddm1*). Data are presented as percentage and were derived from pictures of at least 20 nuclei from three independent experiments and two *ddm1 clf* lines. **(C)** Representative pictures of DAPI staining and H3K9me2 immunodetection of nuclei extracted from 50 fixed 10 d-old Arabidopsis seedlings for each genotype.

meta-analysis of DNA methylation and publicly available MNase data obtained for *ddm1* (Lyons & Zilberman, 2017) confirmed that this periodic hypermethylation is driven by specific methylation of non-nucleosomal linker DNA (Fig 5C). Notably, inter-nucleosomal DNA has been shown to be more accessible to DNA methyltransferases than DNA bound to the nucleosome core particle (Lyons & Zilberman, 2017). Therefore, our findings indicate that CLF activity limit the accessibility of DNA methyltransferases to linker DNA in the absence of DDM1.

## Discussion

In this study, we have shown that hundreds of TEs gain H3K27me3 when hypomethylated by *ddm1*. This phenomenon has two

important mechanistic implications, the first one being that DNA methylation can globally exclude H3K27me3. We and others previously showed that loss of DNA methylation induced by another hypomethylation mutant, *met1*, rather than loss of H3K9me2, allows for H3K27me3 deposition at TEs (Mathieu et al, 2005; Deleris et al, 2012). However, another commonality between *met1* and *ddm1* mutants is the decompaction of their DNA, which could also contribute to favor the access of PRC2 to chromatin, a possibility that could be further investigated by identifying and using genetic backgrounds impaired for chromatin compaction but not DNA methylation. The second mechanistic implication of our observations is that PRC2 can be recruited to TEs. Here, we have characterized the transposons marked by H3K27me3 in *ddm1* and found some motifs recently described in the genic targets of PRC2 and functionally involved in the recruitment of this

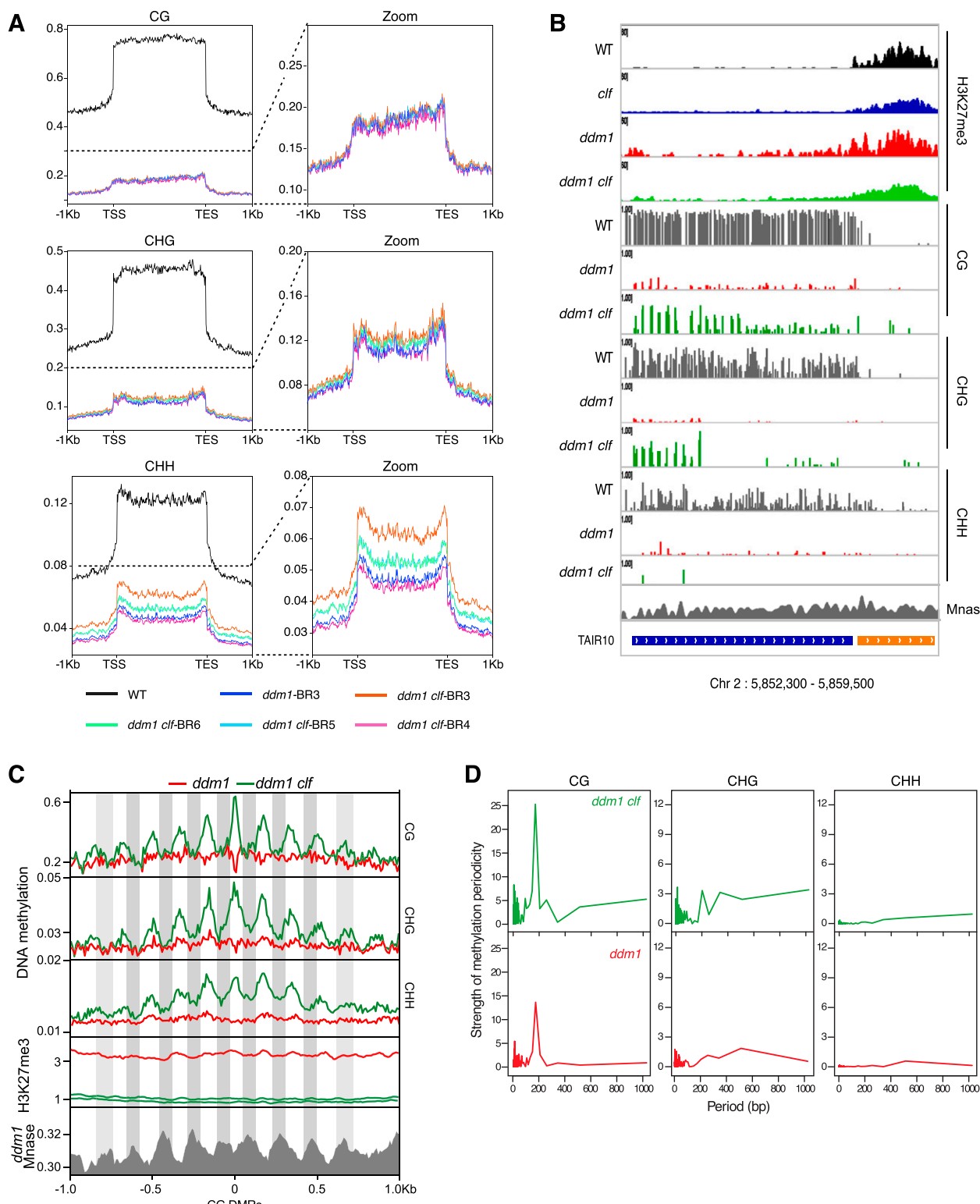

**Figure 5. Periodic hypermethylation is observed in *ddm1 clf* compared with *ddm1*.**
**(A)** Meta-transposable elements (TEs) showing average DNA methylation levels on all TEs, in the three different contexts, in all the genotypes of interest, with a zoom on *ddm1* and *ddm1 clf* genotypes. **(B)** Genome browser view of H3K27me3 and DNA methylation of a TE (blue bar) showing hypermethylation in *ddm1 clf* compared with *ddm1*. **(C)** Plot showing average DNA methylation, H3K27me3, and nucleosome occupancy (every peak in *ddm1* Mnase data show the position of a nucleosome) 1 Kb upstream and downstream of "small" CG differential methylated regions (20 bp) in the three different sequence contexts. **(D)** Fast Fourier transform periogram of average DNA methylation across short-differential methylated regions overlapping well-positioned nucleosomes within TEs for the indicated genotypes. Source data are presented in Table S4.

complex at genes (Xiao et al, 2017). Thus, although some TEs could be marked by H3K27me3 because of spreading from adjacent PcG targets as it is the case for *EVD* (Zervudacki et al, 2018), the presence of these motifs as well as the localization of many TEs far away from any H3K27me3 domain, could be a sequence-based, instructive mode of *cis*-recruitment for PRC2 either through direct motif recognition of the motifs by PRC2 or through the formation of specific chromatin structures in link with the sequence. Alternatively, this recruitment could also be promoted by the enrichment of H2A.Z over demethylated TEs (Zilberman et al, 2008) because a mechanistic link was found between H2A.Z and H3K27me3 (Carter et al, 2018). These non-mutually exclusive scenarios will deserve future investigation.

Subsequently, the compound mutant of DNA methylation and PRC2, *ddm1 clf*, uncovered unexpected, and interesting molecular phenotypes. First, PRC2 targeting of TEs in the *ddm1* background depends almost completely on CLF. This requirement of CLF contrasts with the partial dependency on CLF and SWN at H3K27me3 marked genes, where SWN plays a major role, presumably in nucleating H3K27me3 (Yang et al, 2017). A possible explanation for this result could be that the transcription factors implicated in the nucleation of H3K27me3 (Xiao et al, 2017) at TEs in *ddm1* recruit a FIE–EMF2–PRC2 complex that does not contain SWN.

Second, from a functional point of view and rather unexpectedly, our results did not show that PRC2 acts as a back-up silencing system in the absence of DNA methylation because the loss of H3K27me3 upon CLF mutation did not enhance the number of *ddm1*-activated TEs nor their activation. Instead, molecular phenotypes of partial *ddm1* suppression were observed, which is reminiscent of the observations made in Neurospora whereby a mutation in PRC2 partially rescues some phenotypes caused by a defective H3K9 methylation pathway (Basenko et al, 2015). In fact, in our study, in *ddm1 clf*, we observed that TEs activated in *ddm1* were generally not targeted by PcG and accordingly, they were similarly activated in *ddm1 clf*. But more surprising, the same or lesser number of TEs were transcriptionally active in *ddm1 clf* versus *ddm1*. This can be explained by the partial compaction of DNA observed in *ddm1 clf* compared with *ddm1*, associated with hypermethylation of the DNA in the three sequence contexts. Besides, it is possible that many TEs targeted by PcG in *ddm1* have lost their potential to be transcribed.

One notable exception was the *ATCOPIA93* retroelement *EVD*, which not only was more transcribed in *ddm1-clf* rather than *ddm1* (Zervudacki et al, 2018) but also tended to transpose more in this background (Fig 3). Interestingly, in *ddm1 clf*, *EVD*, and in particular its LTR (which serves as a promoter), did not get remethylated in CG and CHG context, and very slightly in CHH context (Fig S5) which could explain an absence of transcriptional resilencing for this element.

Because the hypermethylation occurred at TEs that are targets of PcG but also non-targets of PcG, we could not conclude on a direct antagonism between H3K27me3 and DNA methylation. As we ruled out obvious scenarios such as expression changes of DNA methylation factors or loss of H1 function in *ddm1 clf*, we can now propose two mechanisms for the global hypermethylation of *ddm1 clf* compared with *ddm1*. First, it is likely that in *ddm1* mutant, like in *met1* (Zilberman et al, 2008), H2AZ is incorporated at TEs. H2AZ is

known to be antagonistic to DNA methylation and given that a mutation of *CLF* leads to loss of H2AZ (Carter et al, 2018), this may favor the reestablishment of DNA methylation. Another non-exclusive possibility could be that the chromatin decompaction displayed by *ddm1* in the presence of PRC2 globally restricts DNA remethylation which would occur as a consequence of DNA recompaction in *ddm1 clf* through unknown mechanisms. TE-derived 24-nt and 21-nt small RNAs, the biogenesis of the later known to be induced in *ddm1* mutants, could participate to either of these processes by directing the corrective reestablishment of TE silencing via the RdDM pathway (Teixeira et al, 2009; Marí-Ordóñez et al, 2013; Nuthikattu et al, 2013). The observation that *EVD*, which produces relatively few siRNAs in comparison to other TEs, is not remethylated supports this hypothesis. Besides, small RNAs were recently shown to be produced upon chromatin decondensation during early embryogenesis or heat-stress and proposed to subsequently help to reconstitute proper heterochromatin (Papareddy et al, 2020).

What favors DNA compaction upon loss of H3K27me3, and, accordingly, what antagonizes chromocenters formation in *ddm1* in the presence of H3K27me3 is also unclear. It could be linked to the replacement of H3.1, the substrate for ATRX5 and 6 and H3K27 monomethylation (a modification associated with chromatin condensation) (Jacob et al, 2009, 2010) by H3.3 and H3K27me3 in *ddm1*, and the subsequent reincorporation of H3.1 in the absence of CLF. In this respect, the *ddm1 clf* mutant represents a valuable system to test the causal and functional relationship between DNA methylation and DNA compaction, which is elusive. As for the possibility of a direct antagonism between H3K27me3 and DNA methylation, we do not exclude it but the use of PcG mutants to analyze the impact of H3K27me3 loss at specific loci currently makes it difficult to tear apart cis-effects of this loss on a given TE from global and indirect effects.

Finally, our work in *ddm1 clf* mutant could have important implications for biological situations where TEs are naturally hypomethylated, DDM1 absent and/or chromatin decompacted. First, in the pollen vegetative nucleus, where hundreds of TEs are DNA-demethylated because of the activity of DEMETER glycosylases (Ibarra et al, 2012), where DDM1 is not detected (Slotkin et al, 2009). TE activation in this cell type is contributed by chromatin decondensation and DNA demethylation–dependent and independent mechanisms (He et al, 2019) and has been proposed to reinforce silencing in the gamete through production of small RNAs that could transit into the sperm cell (Slotkin et al, 2009; Calarco et al, 2012; Ibarra et al, 2012; Martínez et al, 2016). Whether PRC2 targets some of the demethylated TE sequences and whether this could antagonize chromatin decondensation and modulate TE activation and small RNA production in this context is an open question. In this scenario, the mechanisms we have described in this work could play an important role in the reinforcement of silencing in the gametes and its modulation. Similarly, in the endosperm, the nutritive terminal tissue surrounding the seed and derived from the central cell, the chromatin is less condensed than in other types of nuclei (Baroux et al, 2007). In addition, hundreds of TEs of the maternal genome are naturally hypomethylated forming primary imprints and many of these hypomethylated TEs are targeted by PRC2 forming secondary imprints (Weinhofer et al, 2010; Rodrigues & Zilberman, 2015). These secondary imprints cause, for instance, the silencing of

the maternal *PHERES* locus, whereas the cognate paternal allele is activated (Makarevich et al, 2008; Rodrigues & Zilberman, 2015). The interplay between DNA methylation and PRC2 that we have evidenced could thus be particularly relevant in this cell type and modulate the imprinting of some genes as previously suggested (Moreno-Romero et al, 2016). Finally, an exciting possibility is that PRC2 could target transposons after their mobilization and integration and this could slow down the establishment of their DNA methylation-based transgenerational and stable epigenetic silencing.

Our genome-wide analysis brought significant insights into the molecular bases of H3K27me3 deposition at DNA hypomethylated TEs and revealed a global antagonism between H3K27me3 and DNA remethylation. However, the unexpected DNA hypermethylation that we observed in the *ddm1-clf* double mutant versus *ddm1* did not allow us to formally to test our primary hypothesis of enhanced TE activation in the absence of both DNA methylation and H3K27me3 because DNA remethylation may have masked this transient phenomenon. Although studying this double mutant was a necessary step, future studies should involve genetic backgrounds which prevent this global remethylation and compaction of the genome, or alternative systems which enable the specific DNA and H3K27me3 demethylation of discrete loci.

# Materials and Methods

### Plant material and growth condition

All experiments were conducted on seedlings grown on MS plates and with an 8-h light/16-h dark photoperiod, except for Southern blot where analyses were performed on 5-wk-old rosette leaves grown in the same conditions.

### Mutant lines

We used the *ddm1-2* allele (Vongs et al, 1993) and the *clf-29* allele (Bouveret et al, 2006). Double *ddm1 clf* mutants were generated by crossing the abovementioned mutants (Zervudacki et al, 2018). Experiments were performed on F3 progenies using *ddm1-2* second generation plants as controls. We refer to the different progenies tested as different *ddm1-clf* lines or BR throughout the study.

### SDS–PAGE and Western blotting

Chromatin-enriched protein fraction was extracted from 10-d-old seedlings as described in Bourbousse et al (2012), quantified by standard bicinchoninic acid (BCA) assay and 20 µg were resolved on SDS–PAGE. After electroblotting the proteins on a polyvinylidene difluoride (PVDF) membrane, H3K27me3 analysis was performed using an antibody against the H3K27me3 (07-449; Millipore) at a 1:1,000 dilution and a secondary antibody against rabbit coupled to HRP (W4011; Promega) at a 1:20,000 dilution.

### DNA methylation analyses

DNA was extracted using a standard cetyl trimethylammonium bromide (CTAB)-based protocol. Bisulfite conversion, BS-seq libraries, and sequencing (paired-end 100 nt reads) were performed by BGI Tech Solutions. Adapter and low-quality sequences were trimmed using Trimming Galore v0.3.3. Mapping was performed on TAIR10 genome annotation using Bismark v0.14.2 (Krueger & Andrews, 2011) and the parameters: –bowtie2, -N 1, -p 3 (alignment); –ignore 5 –ignore_r2 5 –ignore_3prime_r2 1 (methylation extractor). Only uniquely mapping reads were retained. The methylKit package v0.9.4 (Akalin et al, 2012) was used to calculate differential methylation in 100 or 20 bp non-overlapping windows (DMRs). Significance of calculated differences was determined using Fisher's exact test and Benjamin–Hochberg adjustment of *P*-values (false discovery rate < 0.05) and methylation difference cutoffs of 40% for CG, 20% for CHG, and 20% for CHH. Differentially methylated windows within 100 or 20 bp of each other were merged to form larger DMRs. 100-bp windows with at least six cytosines covered by a minimum of 6 (CG and CHG) and 10 (CHH) reads in all libraries were considered.

DNA methylation periodicity was analyzed by extracting row DNA methylation around (1 Kb) CG short-hyper DMRs between *ddm1-clf* and *ddm1*. Average DNA methylation around (up to 1 Kb) the center of well-positioned nucleosomes (Lyons & Zilberman, 2017) was calculated and analyzed using fast Fourier transform periodogram included in the spec.pgram function in R, which generated the power spectrum over a range of frequencies. "spec" output was plotted against the inverse of frequency "freq," which corresponds to the distance (bp) to nucleosomes centers.

### RNA extraction and 3'quant-seq (RNA) analyses

Total RNA was extracted using Trizol followed by clean-up on RNeasy Plant Mini Kit columns (Macherey-Nagel). 3'end librairies were prepared using the QuantSeq 3' Fwd library prep kit (Lexogen). Libraries were sequenced to acquire 150 bp-reads on a NextSeq Mid output flow cell (Fasteris). Expression level was calculated by mapping reads using STAR v2.5.3a (Dobin et al, 2013) on the *A. thaliana* reference genome (TAIR10) with the following arguments –outFilterMultimapNmax 50 –outFilterMatchNmin 30 –alignSJoverhangMin 3 –alignIntronMax 10000. Counts were normalized and annotations (genes and TEs) were declared differentially expressed between samples (mutants versus wild type) using DESeq2 (Love et al, 2014).

### Chromatin immunoprecipitation and ChIP-seq analyses

ChIP experiments were conducted in two to four BRs of WT, *ddm1*, *clf*, and *ddm1 clf* (14 d-old seedlings) using an anti-H3K27me3 antibody (07-449; Millipore). For each BR, two IPs were carried out using 80 µg of Arabidopsis chromatin mixed with 5 µg of Drosophila chromatin, as quantified using BiCinchoninic Acid assay (Thermo Fisher Scientific). DNA eluted and purified from the two technical replicates was pooled before library preparation (TruSeq ChIP; Illumina) and sequencing (sequencing single-reads, 1 × 50 bp; Illumina) of the resulting input and IP samples performed by Fasteris.

Reads were mapped using Bowtie2 v2.3.2 (Langmead & Salzberg, 2012) onto TAIR10 *A. thaliana* and *Drosophila melanogaster* (dm6) genomes, using the parameters "very-sensitive." Only one mapping instance was kept from reads mapping multiple times with the same score. Reads with more than one mismatch were eliminated.

Normalization factor (Rx) for each sample using spiked-in *D. melanogaster* chromatin was calculated (Nassrallah et al, 2018) using the following formula Rx = r/Nd_IP, where Nd_IP corresponds to the number of reads mapped on *D. melanogaster* genome in the IP and r corresponds to the percentage of *Drosophila*-derived reads in the input. Genomic regions significantly marked by H3K27me3 were identify using MACS2 (Zhang et al, 2008) and genes or TEs overlapping these regions were obtained using bedtools (Quinlan & Hall, 2010). The number of reads over marked genes or TEs were normalized by applying the normalization factor and differentially marked genes between samples were calculated using DESeq2 (Love et al, 2014).

### TE-sequence capture

TE sequence capture was performed on around 100 seedlings in all cases except *ddm1-clf*#23 NOchl were only 12 seedlings were recovered. Seedlings were grown in plates under control (long-day) conditions and genomic DNA was extracted using the CTAB method (Murray & Thompson, 1980). Libraries were prepared as previously described (Quadrana et al, 2019) using 1 *µ*g of DNA and TruSeq paired-end kit (Illumina) following the manufacturer's instructions. Libraries were then amplified through seven cycles of ligation-mediated PCR using the KAPA HiFi Hot Start Ready Mix and primers AATGATACGGCGACCACCGAGA and CAAGCAGAAGACGGCATACGAG at a final concentration of 2 *µ*M. 1 *µ*g of multiplexed libraries were then subjected to TE-sequence capture exactly as previously reported (Quadrana et al, 2016). Pair-end sequencing was performed using one lane of Illumina NextSeq500 and 75 bp reads. About 42 million pairs were sequenced per library and mapped to the TAIR10 reference genome using Bowtie2 v2.3.2 (Langmead & Salzberg, 2012) with the arguments –mp 13 –rdg 8,5 –rfg 8,5 –very-sensitive. An improved version of SPLITREADER (available at https://github.com/LeanQ/SPLITREADER) was used to detect new TE insertions. Briefly, split-reads as well as discordant reads mapping partially on reference and consensus TE sequences (obtained from RepBase update) were identified, soft clipped and remapped to the TAIR10 reference genome using Bowtie2 (Langmead & Salzberg, 2012). Putative insertions supported by at least one split- and/or discordant-reads at each side of the insertion sites were retained. Insertions spanning centromeric repeats or coordinates spanning the corresponding donor TE sequence were excluded. In addition, putative TE insertions detected in more than one library were excluded to retain only sample-specific TE insertions.

### Southern blot

DNA from 5-wk-old rosette leaves was extracted using a standard CTAB protocol. 1.5 *µ*g of genomic DNA was digested overnight with SSpI restriction enzyme. The digestion was run on a 1% agarose gel, transferred to Hybond N+ membranes, blocked, and washed according to manufacturer instructions (GE Healthcare). Membranes were probed with a PCR product (corresponding to a fragment of *EVD* GAG sequence), radiolabeled with *α* 32P-dCTP using the Megaprime DNA Labeling System. EVD PCR product was generated with the same primers as in Marí-Ordóñez et al (2013).

### Pyrosequencing

*ATCOPIA93* DNA from genomic DNA (same extraction as used in Southern Blot) was analyzed as in Zervudacki et al (2018).

### Cytology

DAPI staining on fixed nuclei was performed as described in (Bourbousse et al, 2015) using 10-d-old chopped cotyledons to avoid developmental defects due to accumulation of transposition. For immunodetection, a primary antibody against H3K9me2 (1220; Abcam) at a 1:200 dilution was used, followed by a secondary antibody (488 mouse; Alexa Fluor) at a 1:400 dilution and DAPI staining. Images were acquired with a confocal laser scanning microscope and processed using ImageJ (https://imagej.nih.gov/ij/).

### Motif detection

PRE-like detection in TEs with a gain of H3K27me3 was made using the FIMO tool from http://meme-suite.org with standards parameters.

## Data Availability

Source data used for the figures are presented in Tables S1–S4. Sequencing data has been deposited in the European Nucleotide Archive under project PRJEB34363.

## Supplementary Information

## Acknowledgements

We thank our colleagues for discussions, the Genomics and Informatics facilities at IBENS, the Barneche lab (C Bourbousse and G Teano) for their help with Drosophila spike-in ChIP experiments and A Mari-Ordonez and S Oberlin for discussions on 3'Quant-seq RNA analyses. This work was funded by a Human Frontier Scientific Program Career Development Award (HFSP-CDA-00018/2014) granted to A Deleris with a contribution of the Centre National de la Recherche Scientifique (CNRS) Momentum program granted to L Quadrana for transposon capture. Additional support was from the CNRS.

### Author Contributions

M Rougée: formal analysis, validation, investigation, visualization, methodology, and writing—original draft.
L Quadrana: formal analysis, investigation, and methodology.
J Zervudacki: investigation and methodology.
V Hure: formal analysis, validation, and investigation.
V Colot: resources.
L Navarro: resources.
A Deleris: conceptualization, resources, formal analysis, supervision, funding acquisition, validation, investigation, visualization,

methodology, project administration, and writing—original draft, review, and editing.

## Conflict of Interest Statement

The authors declare that they have no conflict of interest.

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
