## [Reviewer comments · Life Science Alliance]

Life Science Alliance

Polycomb mutant partially suppresses DNA hypomethylation-associated phenotypes in Arabidopsis

Martin Rougée, Leandro Quadrana, Jérôme Zervudacki, Valentin Hure, Vincent Colot, Lionel Navarro, and Angélique Deleris

DOI: <https://doi.org/10.26508/lsa.202000848>

Corresponding author(s): Angélique Deleris, Institut de Biologie Intégrative de la Cellule (I2BC)

Review Timeline:

Submission Date:	2020-07-13
Editorial Decision:	2020-08-13
Revision Received:	2020-10-21
Editorial Decision:	2020-11-15
Revision Received:	2020-11-25
Accepted:	2020-12-01

Scientific Editor: Shachi Bhatt

Transaction Report:

August 13, 2020

Re: Life Science Alliance manuscript #LSA-2020-00848

Dr. Angélique Deleris
Institut de biologie de l'Ecole normale supérieure (IBENS)/ Institut de Biologie Intégrative de la
Cellule (I2BC)
Ecology and Evolutionary Biology/ Genome Biology
46 rue d'Ulm /Rue de la Terrasse
PARIS/ GIF-SUR-YVETTE 75005/ 91190
France

Dear Dr. Deleris,

Thank you for submitting your manuscript entitled "Polycomb mutant partially suppresses DNA hypomethylation-associated phenotypes in Arabidopsis" to Life Science Alliance. The manuscript was assessed by expert reviewers, whose comments are appended to this letter.

As you will note, all reviewers are quite enthusiastic about the findings from this study, and have asked for addressable minor points for the revision. We encourage you to revise the manuscript in accordance to the reviewers' points, and re-submit a revised manuscript that addresses all the concerns raised by the reviewers.

Thank you for this interesting contribution to Life Science Alliance. We are looking forward to receiving your revised manuscript.

Sincerely,

Shachi Bhatt
Executive Editor
Life Science Alliance

B. MANUSCRIPT ORGANIZATION AND FORMATTING:

Reviewer #1 (Comments to the Authors (Required)):

This is an excellent study examining the interaction of DNA methylation and PRC2 regulation of TEs in the Arabidopsis genome. The work complements understanding from lower organisms where H3K27me3 also redistributes to TEs. The data are complex, but the authors do a good job of interpretation and discussion. A few minor points:

Fig.2 - a ddm1 swm double mutant needs to be included in the analysis if they want to conclude that K27me3 enrichment at TEs is fully dependent on CLF.

Line 116+ they argue that H3K27me3 in ddm1 is redistributed, not ectopically gained, but there is

no decrease in genic H3K27me3 enrichment. They reason that this is caused by low number of targeted TEs in *ddm1* (in contrast to *met1*) - suggesting the H3K27me3 quantification is not sensitive enough. More discussion is required on the discrepancy between *met1* vs *ddm1* and the numbers of genes that lose K27me3 in *met1* (S1A shows a comparison only for TEs). Line 268 they conclude there is a sequence-based, instructive mode of cis-recruitment for PRC2. This conclusion needs to be tempered as short enriched sequence motifs may just promote chromatin structures, which recruit PRC2.

Reviewer #2 (Comments to the Authors (Required)):

The authors previously showed, in an influential paper, that loss of DNA methylation caused by mutation of the maintenance DNA methyltransferase *met1* is associated with gain of the PRC2 mark H3K27me3 in transposable elements in *Arabidopsis* (Deleris et al., Plos Genet, 2012). DDM1 is a chromatin remodeler whose mutation induces DNA hypomethylation in heterochromatin. One reasonable hypothesis based on the previous findings and on their more recent work on the retrotransposon EVADE, is that H3K27me3 is compensating for DNA methylation. If this hypothesis is correct, additional loss of H3K27me3 should cause more severe *ddm1* phenotypes. Thus, it is surprising that this manuscript shows that a *ddm1* *clf* double mutant has less TE transcription and transposition than *ddm1*, even at those TEs that gain H3K27me3 in *ddm1* mutants. Rather, the behavior of EVADE does not apply to most other TEs. These findings are of broad interest to the plant epigenetics community and suitable for publication in LSA. The main points of the paper are strongly supported.

I have a few comments for improving the manuscript.

1. While the 140 bp periodic hypermethylation in *ddm1* *clf* is potentially interesting from a mechanistic perspective (Fig S5A), the authors should confirm this methylation pattern by locus specific bisulfite-PCR. This looks like it could be a read mapping artifact.
2. Is the gain of H3K27me3 specific to TEs in *ddm1*? In the browser snapshot in Fig 1A (right side) it appears that orange regions (genes) also gain H3K27me3.
3. In the discussion section, three hypotheses are put forth to explain the phenotype. One is that chromatin decompaction in *ddm1* prevents DNA remethylation (line 294-296). Yet it has been shown that RdDM targets decompacted chromatin (Schoft et al., EMBO Rep, 2009), which would argue against this possibility. Their discussion of small RNAs in this section was confusing.
4. Indicate the tissue used for ChIP. Please also provide the read alignment parameters for mapping ChIP reads with Bowtie2.
5. For the data in Figure 4, indicate whether types (1, 2, or 3) were blind-assigned to the imaged nuclei (i.e. by someone unaware of the genotype of the nuclei they were looking at).
6. In Figure 1F, the enriched sequence motifs are not shown to be PREs in this study, but are hypothesized as such. Relabel figure accordingly as 'Possible PREs' instead.

Reviewer #3 (Comments to the Authors (Required)):

This paper follows up on a previous intriguing observation that many transposons gain the polycomb silencing mark H3K27me3 in DNA methylation mutants. In this work they show that in

ddm1 100s or TEs gain H3K27me3, which they show is dependent on CURLY LEAF.

Previously the authors showed gain of K27me3 on TEs in the met1 mutant, which is here extended to ddm1 - the ddm1 mutant has a more specific effect on repeat sequences than met1. They observe that 672 TEs, mainly located in the pericentromeres, gain H3K27me3 in ddm1.

Please state what the overlap of these elements is with those previously shown to gain H3K27me3 in met1 in the main text - this is found in Fig S1A - I think this could be good to include in a main figure? It's clear that the extent of gain of H3K27me3 in ddm1 is relatively less than in met1 - do the authors have an explanation for this? It would be interesting in Fig. S1B to compare the family profile for ddm1 and met1. Fig. S1C - how do these specific regions compare in met1? I think generally the met1 vs ddm1 comparison could be made in more detail throughout the paper. Another interesting difference is that genes lost H3K27me3 in met1 but not ddm1.

The authors test for DNA motifs enriched in the TEs that gain H3K27me3 and observe enrichment of some short motifs. The study from Xiao et al 2017 identifies some targeting motifs - have the authors looked for these specifically?

The authors show that in ddm1 clf1 double mutants gain of H3K27me3 is lost on TEs, and interestingly the presence of SWN was not able to compensate.

Due to ddm1 clf1 showing pleiotropic phenotypes the authors tested whether TE mobilization played a role. They focused on EVADE and ATR which are COPIA93 elements. Interestingly, many new insertions are found, to a greater degree than seen in ddm1 alone, implying that the H3K27me3 is repressing these TEs in the absence of DDM1.

In ddm1 about a 1000 TEs are upregulated at the RNA level. However, most of these did not gain H3K27me3. Surprisingly, the general picture was different from EVADE, with most TEs not showing enhanced expression or movement in ddm1 clf1. Why do they authors think that EVADE and ATR behave differently to the general picture?

Could the authors show genome screenshots of EVADE and ATR showing H3K27me3 profiles and RNA-seq data in all the genotypes that analysed?

The authors look at chromocenter morphology using DAPI staining and H3K9me2 IF. As reported ddm1 shows a loss of chromocentres, which is partially suppressed in ddm1 clf. Surprisingly, the ddm1 nuclei still show appreciable H3K9me3 staining - is this expected? I was under the impression that this mark is absent in ddm1?

One of the most interesting findings in the paper is that in ddm1 and to a greater extent in ddm1 clf DNA methylation shows a marked periodicity that is approximately nucleosomal, which appears to be on the linker regions.

Figure 1A. Please label clearly that this data represents H3K27me3 enrichment. Please also add a key to explain what the blue vs orange annotation units are.

Figure 2B and 2C - please add wild type to these plots.

Minor points.

Line 49 - I believe DRM2 also plays a minor role in maintenance?

Line 52 - please provide references for this statement.

Line 73 - spell out EVADE as this is the first mention.

Title of Figure 3 - I would write this as 'Transposon activation in ddm1 and ddm1 clf'.

We thank the reviewers for constructive critics that we have addressed to the best of our ability and which have definitely helped us to improve the manuscript.

Editor/reviewer comments are below in **bold type** and our response is in regular type.

Reviewer #1 (Comments to the Authors (Required)):

This is an excellent study examining the interaction of DNA methylation and PRC2 regulation of TEs in the Arabidopsis genome. The work complements understanding from lower organisms where H3K27me3 also redistributes to TEs. The data are complex, but the authors do a good job of interpretation and discussion. A few minor points:

Fig.2 - a *ddm1 swn* double mutant needs to be included in the analysis if they want to conclude that K27me3 enrichment at TEs is fully dependent on CLF.

We have now isolated a *ddm1 swn* double mutant which we analyzed for H3K27me3 levels, by ChIP-qPCR, at five representative TEs (diverse locations in the genome) that gain H3K27me3 in *ddm1* to various extents, and lose it completely in *ddm1 clf* according to our ChIP seq data. For all five loci, we observed no significant loss of H3K27me3 in *ddm1 swn* compared to *ddm1*. We have now included these data in the Figure 2 and added the following sentence (in bold) in the manuscript:

*Conversely, the gain of H3K27me3 observed over TEs in *ddm1* was almost completely abolished in *ddm1 clf* (Fig 2A-C and S2C Fig) while being unchanged at all TEs tested in *ddm1 swn* (Fig 2D, S2D Fig). Together, these results show that deposition of H3K27me3 at most TEs in *ddm1* is fully dependent on CLF with no apparent role of the paralogous histone methyltransferase SWN.*

Line 116+ they argue that H3K27me3 in *ddm1* is redistributed, not ectopically gained, but there is no decrease in genic H3K27me3 enrichment. They reason that this is caused by low number of targeted TEs in *ddm1* (in contrast to *met1*) - suggesting the H3K27me3 quantification is not sensitive enough. More discussion is required on the discrepancy between *met1* vs *ddm1* and the numbers of genes that lose K27me3 in *met1* (S1A shows a comparison only for TEs).

We agree with the reviewer that there is no evidence for redistribution of H3K27me3 and “redistribution” has now been replaced by “ectopic gain” in the text.

A metagene of H3K27me3 is now shown in Fig S1D showing that there is no loss of H3K27me3 in *ddm1* versus WT at genes (supporting our previous statement “0 gene was found to lose H3K27me3 in *ddm1*”).

In addition, we have now further discussed the discrepancy between *ddm1* and *met1* as for loss of genic H3K27me3 in the light of the new analyses that we generated and presented in Figure S1E (DNA methylation profiles across genes in *ddm1* versus WT). The text has been modified as follows:

*Of note, ectopic gain of H3K27me3 to TEs in *ddm1* did not seem associated with a loss of H3K27me3 at genes (Fig S1D, and no gene lost H3K27me3 significantly in *ddm1* in our differential analysis) contrary to what was observed in *met1* (Deleris et al, 2012). In the scenario whereby the gain of H3K27me3 at TEs would be the result of redistribution from genes to TEs, this could be explained by the lesser number of TEs targeted by PRC2 in *ddm1* versus *met1* (Fig 1F) presumably because a lesser number of TEs become hypomethylated in *ddm1* (heterochromatic TEs mostly). Alternatively, or in addition, loss of H3K27me3 at genes in *met1* but not in *ddm1* could be contributed by the pronounced ectopic DNA hypermethylation at many genes, particularly H3K27me3-marked genes in *met1* (Deleris et al., 2012) which we did not detect globally in *ddm1* (S1E Fig) even if this phenomenon could occur sporadically at specific genic loci like AGAMOUS (Jacobsen et al., 2000).*

Line 268 they conclude there is a sequence-based, instructive mode of cis-recruitment for PRC2. This conclusion needs to be tempered as short enriched sequence motifs may just promote chromatin structures, which recruit PRC2.

We agree with the reviewer: our conclusion has now been tempered and this alternative possibility has now been suggested in the text.

*These results **suggest** the presence of an instructive mechanism of PRC2 recruitment at TEs with particular motifs used as nucleation sites **either through direct sequence recognition or, indirectly, through chromatin structures that could be promoted by these sequences.***

Reviewer #2 (Comments to the Authors (Required)):

The authors previously showed, in an influential paper, that loss of DNA methylation caused by mutation of the maintenance DNA methyltransferase *met1* is associated with gain of the PRC2 mark H3K27me3 in transposable elements in Arabidopsis (Deleris et al., Plos Genet, 2012). DDM1 is a chromatin remodeler whose mutation induces DNA hypomethylation in heterochromatin. One reasonable hypothesis based on the previous findings and on their more recent work on the retrotransposon EVADE, is that H3K27me3 is compensating for DNA methylation. If this hypothesis is correct, additional loss of H3K27me3 should cause more severe *ddm1* phenotypes. Thus, it is surprising that this manuscript shows that a *ddm1 clf* double mutant has less TE transcription and transposition than *ddm1*, even at those TEs that gain H3K27me3 in *ddm1* mutants. Rather, the behavior of EVADE does not apply to most other TEs. These findings are of broad interest to the plant epigenetics community and suitable for publication in LSA. The main points of the paper are strongly supported.

I have a few comments for improving the manuscript.

1. While the 140 bp periodic hypermethylation in *ddm1 clf* is potentially interesting from a mechanistic perspective (Fig S5A), the authors should confirm this methylation pattern by locus specific bisulfite-PCR. This looks like it could be a read mapping artifact.

We reasoned that if this had been a mapping artifact, it would have been the case in all the genetic backgrounds tested, which is not the case. In addition, this peculiar pattern had been previously reported specifically in another mutant—*ddm1 h1*—in a well-executed study performed by DNA methylation specialists (Zemach et al., 2013; Lyons et al., 2017) which we now mention in the last section of the results. We thus believe that additional locus specific bisulfite-PCR are not absolutely necessary. However, if the reviewer is not convinced by our arguments, we would be happy to discuss further what he/she had in mind.

2. Is the gain of H3K27me3 specific to TEs in *ddm1*? In the browser snapshot in Fig 1A (right side) it appears that orange regions (genes) also gain H3K27me3.

Yes the gain of H3K27me3 is specific to TEs since we found that globally there is no gain (or loss) of H3K27me3 in *ddm1* at genes as shown by the metagene presented in S2B Fig (also shown now in Fig S1D as requested by another referee).

Thus, for clarity, we did not comment on sporadic examples like the (minor) gain in Figure 1A which could be the result of spreading from nearby TEs; alternatively, they could be “TE-like”

pseudogenes that gain H3K27me3 as previously observed in *met1* (see Deleris et al., 2012, Table S3).

3. In the discussion section, three hypotheses are put forth to explain the phenotype. One is that chromatin decompaction in *ddm1* prevents DNA remethylation (line 294-296). Yet it has been shown that RdDM targets decompacted chromatin (Schoft et al., EMBO Rep, 2009), which would argue against this possibility. Their discussion of small RNAs in this section was confusing.

We thank the reviewer for pointing that these points could be confusing for the reader and have tried to clarify them.

Small RNAs have been shown to be produced in response to DNA demethylation and/or chromatin decondensation to subsequently promote DNA-remethylation (as shown by Teixeira et al, 2009; Nuthikattu et al, 2013; Marí-Ordóñez et al, 2013) and/or chromatin recompaction (as proposed in Papareddy et al, 2020). We have now tried to make this clearer by adding the last sentence in bold in the paragraph below.

*TE-derived 24-nt and 21-nt small RNAs, the biogenesis of the later known to be induced in *ddm1* mutants, could participate to either of these processes by directing the corrective reestablishment of TE silencing via the RdDM pathway (Teixeira et al, 2009; Nuthikattu et al, 2013; Marí-Ordóñez et al, 2013). The observation that EVD, which produces relatively few siRNAs in comparison to other TEs, is not remethylated, supports this hypothesis. **Besides, small RNAs were recently shown to be produced upon chromatin decondensation during early embryogenesis or heat-stress and proposed to subsequently help to reconstitute proper heterochromatin (Papareddy et al, 2020).***

4. Indicate the tissue used for ChIP. Please also provide the read alignment parameters for mapping ChIP reads with Bowtie2.

These informations have now been added to the manuscript in the material and methods section.

5. For the data in Figure 4, indicate whether types (1, 2, or 3) were blind-assigned to the imaged nuclei (i.e. by someone unaware of the genotype of the nuclei they were looking at).

The analysis was done blindly and this is now stated.

6. In Figure 1F, the enriched sequence motifs are not shown to be PREs in this study, but are hypothesized as such. Relabel figure accordingly as 'Possible PREs' instead.

This correction has now been added on the corresponding figure (now Fig 1G).

Reviewer #3 (Comments to the Authors (Required)):

This paper follows up on a previous intriguing observation that many transposons gain the polycomb silencing mark H3K27me3 in DNA methylation mutants. In this work they show that in *ddm1* 100s or TEs gain H3K27me3, which they show is dependent on CURLY LEAF.

Previously the authors showed gain of K27me3 on TEs in the *met1* mutant, which is here extended to *ddm1* - the *ddm1* mutant has a more specific effect on repeat sequences than *met1*. They observe that 672 TEs, mainly located in the pericentromeres, gain H3K27me3 in *ddm1*.

Please state what the overlap of these elements is with those previously shown to gain H3K27me3 in *met1* in the main text - this is found in Fig S1A - I think this could be good to include in a main figure?

We agree with the reviewer: this correction has been done in the text and FigS1A is now **Fig1F**.

Its clear that the extent of gain of H3K27me3 in *ddm1* is relatively less than in *met1* - do the authors have an explanation for this?

We have now further discussed the discrepancy between *ddm1* and *met1*. The text has been modified as follows (addition in **bold**):

Moreover, the vast majority of those 672 TEs are located in pericentromeric regions (Fig 1E) and were included in the subset of TEs that gain H3K27me3 in met1 (Deleris et al, 2012) (Fig 1F) possibly because the extent of TE hypomethylation in ddm1 is less than in met1, where all CG methylation (the most abundant) is virtually eliminated (Stroud et al, 2013). This argues for a major role of DNA methylation (in particular CG methylation rather than non-CG methylation associated with H3K9me2) in antagonizing PRC2, as previously proposed (Mathieu et al, 2005; Deleris et al, 2012).

Besides, the difference in the number of TEs that gain H3K27me3 between *met1* and *ddm1* could also be contributed by differences in the methods and differential analyses since ChIP-CHIP was employed for *met1* H3K27me3 analysis (Deleris et al., 2012) versus ChIP-seq for *ddm1* (the present study). This is now stated in the figure legend so that the reader can also take this parameter into consideration.

It would be interesting in Fig. S1B to compare the family profile for *ddm1* and *met1*. Fig. S1C how do these specific regions compare in *met1*?

The distribution of TEs that gain H3K27me3 in *met1* is now presented in FigS1B and the comparison between the TEs that gain H3K27me3 in *ddm1* and *met1* made and discussed in the text as follows:

Two major TE super families (LTR/Gypsy, DNA/others) were overrepresented among the 672 TEs that significantly gain H3K27me3 in ddm1 as compared to the distribution of the heterochromatic, pericentromeric TEs (targets of DDM1) families (S1B Fig). The differences in TE-type targeting between met1 and ddm1 (S1B Fig) likely reflect a differential sensitivity of the TE families to the different mutation as for their DNA methylation, thus the differential extent of TE hypomethylation and loss of PRC2 antagonism by DNA methylation as discussed earlier. In addition, the over-representation of two TE families among the TEs that gain H3K27me3, common to both mutants (S1B Fig) could suggest the existence of sequence-specific targeting.

I think generally the *met1* vs *ddm1* comparison could be made in more detail throughout the paper. Another interesting difference is that genes lost H3K27me3 in *met1* but not *ddm1*.

We agree with this and have now discussed this aspect further, as also requested by referee 1.

A metagene of H3K27me3 is now shown in Fig S1D further showing that there is no loss of H3K27me3 in *ddm1* versus WT at genes (supporting our previous statement "O gene was found to lose H3K27me3 in *ddm1*").

In addition, we have now further discussed the discrepancy between *ddm1* and *met1* as for loss of genic H3K27me3 in the light of the new analyses that we generated and presented in

Figure S1E (DNA methylation profiles across genes in *ddm1* versus WT). The text has been modified as follows:

Of note, ectopic gain of H3K27me3 to TEs in ddm1 did not seem associated with a loss of H3K27me3 at genes (S1D Fig) and no gene lost H3K27me3 significantly in ddm1 in our differential analysis, contrary to what was observed in met1 (Deleris et al, 2012). In the scenario whereby the gain of H3K27me3 at TEs would be the result of redistribution from genes to TEs, this could be explained by the lesser number of TEs targeted by PRC2 in ddm1 versus met1 (Fig 1F) presumably because a lesser number of TEs become hypomethylated in ddm1 (heterochromatic TEs mostly). Alternatively, or in addition, loss of H3K27me3 at genes in met1 but not in ddm1 could be contributed by the pronounced ectopic DNA hypermethylation at many genes, particularly H3K27me3-marked genes in met1 (Deleris et al., 2012) which we did not detect globally in ddm1 (S1E Fig) even if this phenomenon could occur sporadically at specific genic loci like AGAMOUS (Jacobsen et al., 2000).

The authors test for DNA motifs enriched in the TEs that gain H3K27me3 and observe enrichment of some short motifs. The study from Xiao et al 2017 identifies some targeting motifs - have the authors looked for these specifically?

Yes we looked for these particular published motifs and this has now been clarified in the legend.

The authors show that in *ddm1 clf1* double mutants gain of H3K27me3 is lost on TEs, and interestingly the presence of SWN was not able to compensate.

Due to *ddm1 clf1* showing pleiotropic phenotypes the authors tested whether TE mobilization played a role. They focused on EVADE and ATR which are COPIA93 elements. Interestingly, many new insertions are found, to a greater degree than seen in *ddm1* alone, implying that the H3K27me3 is repressing these TEs in the absence of DDM1.

In *ddm1* about a 1000 TEs are upregulated at the RNA level. However, most of these did not gain H3K27me3. Surprisingly, the general picture was different from EVADE, with most TEs not showing enhanced expression or movement in *ddm1 clf1*. Why do they authors think that EVADE and ATR behave differently to the general picture?

We have now proposed an explanation for this in the discussion section, as follows:

*One notable exception was the ATCOPIA93 retroelement EVD, which not only was more transcribed in *ddm1-clf* rather than *ddm1* (Zervudacki et al, 2018) but also tended to transpose more in this background (Fig. 3). Interestingly, in *ddm1 clf*, EVD, and in particular its LTR (which serves as a promoter), did not get remethylated in CG and CHG context, and very slightly in CHH context (S5D Fig) which could explain an absence of transcriptional resiliencing for this element.*

...

We have further proposed an explanation as for why EVD does not get remethylated in *ddm1 clf*:

*TE-derived 24-nt and 21-nt small RNAs, the biogenesis of the later known to be induced in *ddm1* mutations, could participate to either of these processes by directing the corrective reestablishment of TE silencing via the RdDM pathway (Teixeira et al, 2009; Nuthikattu et al, 2013; Mari-Ordóñez et al, 2013). The observation that EVD, which produces relatively few siRNAs in comparison to other TEs, is not remethylated is in support of this hypothesis.*

Could the authors show genome screenshots of EVADE and ATR showing H2K27me3 profiles and RNA-seq data in all the genotypes that analysed?

These screenshots are now shown in **S5D Fig**.

The authors look at chromocenter morphology using DAPI staining and H3K9me2 IF. As reported *ddm1* shows a loss of chromocentres, which is partially suppressed in *ddm1 clf*. Surprisingly, the *ddm1* nuclei still show appreciable H3K9me2 staining - is this expected? I was under the impression that this mark is absent in *ddm1*?

Yes this was not surprising since CHG methylation is not virtually eliminated in the *ddm1* mutant (Stroud et al., 2013) hence the SUVH/CMT3 mechanistic reinforcement loop is still active and H3K9me2 marks can be detected.

Since this was expected, we chose not to comment on it for fluidity of the text.

One of the most interesting findings in the paper is that in *ddm1* and to a greater extent in *ddm1 clf* DNA methylation shows a marked periodicity that is approximately nucleosomal, which appears to be on the linker regions.

Figure 1A. Please label clearly that this data represents H3K27me3 enrichment. Please also add a key to explain what the blue vs orange annotation units are.

This has now been corrected.

Figure 2B and 2C - please add wild type to these plots.

In the second set of ChIP experiments, we didn't use the wild type Col but instead the genetically wild type "WT" coming from the cross *ddm1x clf* and which we genotyped in the F2 as DDM1^{+/+} CLF^{+/+} (likewise in this experiment "*ddm1*" is DDM1^{-/-} CLF^{+/+} and "*ddm1 clf*" is DDM1^{-/-} CLF^{-/-}).

As a result, in this "WT", there are DNA hypomethylated TE which do not get immediately remethylated after reintroducing the wild-type allele of DDM1 and which segregate in the F2, as previously described in the *ddm1*-derived EpiRILs (Teixeira et al, 2009, Cortijo et al, 2014). These stably hypomethylated, segregating TEs are thus prone to be targeted by PRC2 and marked by H3K27me3, even if DDM1 is not mutated in this background. This is exactly what we observed: the average levels of H3K27me3 in this "WT" were intermediate between the "real" (genetically and epigenetically) wild-type Col and *ddm1* mutant.

This is an interesting and predictable observation but we chose not to show it for clarity, since the point of this figure is to show the complete loss of H3K27me3 in *ddm1 clf* versus *ddm1*. We thought that showing this WT entails the explanation of complicated genetics/epigenetics as above that are unnecessary for the readers and could distract them from the main message. We hope the referee agrees with our judgement but if not, we are willing to discuss. The complete figure can be found below for him/her to see.

Minor points.

Line 49 - I believe DRM2 also plays a minor role in maintenance?

This has now been explicated in the next line, with the addition of DRM2 in front of RdDM, as follows:

Maintenance of DNA methylation over TEs is achieved by the combined and context-specific action of DRM2-RdDM (CHH methylation), CHROMOMETHYLASES 2 and 3 (CMT2 and CMT3, for CHH and CHG methylation, respectively) (Zemach et al, 2013; Stroud et al, 2014) and METHYLTRANSFERASE1 (MET1) (CG methylation) (Kankel et al, 2003).

Line 52 - please provide references for this statement.

References have now been added

Line 73 - spell out EVADE as this is the first mention.

Title of Figure 3 - I would write this as 'Transposon activation in *ddm1* and *ddm1 clf*'.

These two points have been corrected.

Of note, we had already ruled out in the first version of the manuscript that the phenotype of DNA hypermethylation in *ddm1 clf* versus *ddm1* was due to expression changes for the components involved in DNA methylation in *ddm1 clf* (**S5B Fig**). During the revision process, as part of our efforts to understand this molecular phenotype, we also ruled out the possibility that this phenomenon was due to a histone H1 loss-of-function in the *ddm1 clf* background. The rationale for testing this was that the DNA methylation pattern in *ddm1 clf* was reminiscent of the one observed in *ddm1 h1* (Zemach et al, 2013; Lyons & Zilberman, 2017). We have now included data supporting this statement (**S5C Fig**) and added the following sentence in the text (after the description of S5B Fig) as follows:

*We did not find any consistent expression changes for the components involved in DNA methylation in *ddm1 clf* (**S5B Fig**), thus these observations cannot be explained, even partially, by the impact of *ddm1 clf* double mutation on the transcriptome. Besides, even if similar patterns of DNA hypermethylation compared to *ddm1* were previously observed in *ddm1 h1* where both *DDM1* and canonical linker histone genes *H1.1* and *H1.2* are mutated (Zemach et al, 2013; Lyons & Zilberman, 2017), *ddm1 h1* mutant does not phenocopy the *ddm1 clf* mutant with regards to chromocenter formation: in fact, contrary to *ddm1 clf*, *ddm1 h1* did not induce DNA recompaction (**S5C Fig**) in agreement with H1 role in chromatin condensation (He et al, 2019). Thus, the DNA hypermethylation observed in *ddm1 clf* versus *ddm1* cannot be attributed either to a histone H1 loss-of-function in this genetic background.*

November 15, 2020

RE: Life Science Alliance Manuscript #LSA-2020-00848R

Dr. Angélique Deleris
Institut de Biologie Intégrative de la Cellule (I2BC)
Ecology and Evolutionary Biology/ Genome Biology
46 rue d'Ulm /Rue de la Terrasse
PARIS/ GIF-SUR-YVETTE 75005/ 91190
France

Dear Dr. Deleris,

Thank you for submitting your revised manuscript entitled "Polycomb mutant partially suppresses DNA hypomethylation-associated phenotypes in Arabidopsis". We would be happy to publish your paper in Life Science Alliance pending final revisions necessary to meet our formatting guidelines.

Along with the points listed below, please also attend to the following,

- please add Author Contributions to your main manuscript text
- please use the [10 author names, et al.] format in your references (i.e. limit the author names to the first 10)
- please add your table legends to the main manuscript text and upload your Table S4 as a separate file
- please add a callout for Figure S1A in your main manuscript text
- please double check your supplementary figure legends; numbering goes from Fig S3 to Fig S5; please upload your Figure S4 or adjust figure legends (currently missing Figure S4)
- please note that figures must fit on one page; your Fig S5 currently spans 2 pages; if you need to split this into 2 supplementary figures, that is perfectly fine
- We would also encourage you to change the figures used in 4B, as currently they are the same as some of the panels in 4A. While we understand that the images in 4B are just meant as a representation to help understand the graph, we do encourage all authors to not repeat the figure panels within or across figures, in general. If you still think that using the same panels is necessary, we would request you to make that clear in the figure legend.

A. FINAL FILES:

B. MANUSCRIPT ORGANIZATION AND FORMATTING:

Sincerely,

Shachi Bhatt, Ph.D.

Executive Editor
Life Science Alliance
<https://www.lsjournal.org/>
Tweet @SciBhatt @LSAJournal

Reviewer #1 (Comments to the Authors (Required)):

The authors have done an excellent job at revising the manuscript in response to the reviewer comments.

Reviewer #2 (Comments to the Authors (Required)):

The authors have satisfactorily addressed all of my comments. I recommend publication without further revision.

Reviewer #3 (Comments to the Authors (Required)):

Thank you - the authors have fully addressed my comments. This is an excellent paper that will be of great interest to the field.

December 1, 2020

RE: Life Science Alliance Manuscript #LSA-2020-00848RR

Dr. Angélique Deleris
Institut de Biologie Intégrative de la Cellule (I2BC)
Genome Biology
Avenue de la Terrasse
GIF-SUR-YVETTE 91190
France

Dear Dr. Deleris,

Thank you for submitting your Research Article entitled "Polycomb mutant partially suppresses DNA hypomethylation-associated phenotypes in Arabidopsis". It is a pleasure to let you know that your manuscript is now accepted for publication in Life Science Alliance. Congratulations on this interesting work.

DISTRIBUTION OF MATERIALS:

Again, congratulations on a very nice paper. I hope you found the review process to be constructive and are pleased with how the manuscript was handled editorially. We look forward to future exciting submissions from your lab.

Sincerely,

Shachi Bhatt, Ph.D.

Executive Editor

Life Science Alliance

<https://www.lsjournal.org/>
